# Emission Characteristics and Health Risks of Volatile Organic Compounds (VOCs) Measured in a Typical Recycled Rubber Plant in China

**DOI:** 10.3390/ijerph19148753

**Published:** 2022-07-19

**Authors:** Shuang Wang, Yucheng Yan, Xueying Gao, Hefeng Zhang, Yang Cui, Qiusheng He, Yuhang Wang, Xinming Wang

**Affiliations:** 1School of Environmental Science and Engineering, Taiyuan University of Science and Technology, No. 66 Waliu Road Wanbailin District, Taiyuan 030024, China; s20202301014@stu.tyust.edu.cn (S.W.); yanyucheng1021@163.com (Y.Y.); gaoxueying@sxist.edu.cn (X.G.); cuiyang201110@163.com (Y.C.); 2Chinese Research Academy of Environmental Sciences (CRAES), Ministry of Environment Protection (MEP), Beijing 100012, China; zhanghf@craes.org.cn; 3School of Earth and Atmospheric Science, Georgia Institute of Technology, Atlanta, GA 30332, USA; yuhang.wang@eas.gatech.edu; 4State Key Laboratory of Organic Geochemistry, Guangzhou Institute of Geochemistry, Chinese Academy of Sciences, Guangzhou 510640, China; wangxm@gig.ac.cn

**Keywords:** volatile organic compounds, recycled rubber plant, source profiles, health risk assessment, OH radical loss rate

## Abstract

The continued development of the automotive industry has led to a rapid increase in the amount of waste rubber tires, the problem of “black pollution” has become more serious but is often ignored. In this study, the emission characteristics, health risks, and environmental effects of volatile organic compounds (VOCs) from a typical, recycled rubber plant were studied. A total of 15 samples were collected by summa canisters, and 100 VOC species were detected by the GC/MS-FID system. In this study, the total VOCs (TVOCs) concentration ranged from 1000 ± 99 to 19,700 ± 19,000 µg/m^3^, aromatics and alkanes were the predominant components, and m/p-xylene (14.63 ± 4.07%–48.87 ± 3.20%) could be possibly regarded as a VOCs emission marker. We also found that specific similarities and differences in VOCs emission characteristics in each process were affected by raw materials, production conditions, and process equipment. The assessment of health risks showed that devulcanizing and cooling had both non-carcinogenic and carcinogenic risks, yarding had carcinogenic risks, and open training and refining had potential carcinogenic risks. Moreover, m/p-xylene and benzene were the main non-carcinogenic species, while benzene, ethylbenzene, and carbon tetrachloride were the dominant risk compounds. In the evaluation results of L_OH_, m/p-xylene (25.26–67.87%) was identified as the most key individual species and should be prioritized for control. In conclusion, the research results will provide the necessary reference to standardize the measurement method of the VOCs source component spectrum and build a localized source component spectrum.

## 1. Introduction

As important substances participating in atmospheric photochemical reactions, volatile organic compounds (VOCs) are the main precursors of secondary organic aerosols (SOA) and tropospheric ozone(O_3_), which are also the major factor for photochemical smog and haze [1,2,3,4,5,6]. Meanwhile, many substances in VOCs, such as benzene, toluene, and other benzene series, are also carcinogenic, teratogenic, and mutagenic and can directly enter the human body through the respiratory tract, skin, and other routes, seriously affecting human health [7,8,9,10]. With the rapid development of urban industrialization in the country, the increase in human activities has led to significantly higher anthropogenic VOC emissions in most areas [11,12]. Furthermore, industrial VOCs account for more than 50% of anthropogenic VOC emissions [13] and have become the key to VOC emission reduction in air pollution prevention and control [14,15,16].

Previous studies on the emission characteristics of VOCs from industrial sources at home and abroad have mainly focused on industries such as the chemical industry, oil refineries, and solvent use. Dumanoglu et al. [17] analyzed the emission sources of Turkey’s heavy industry region and proposed that refineries and petroleum products, petrochemical industry, solvent use, industrial processes, and vehicle exhaust gas were identified as VOC sources, accounting for 56%, 22%, 12%, and 10%, respectively; a study by Saeaw et al. [18] on industrial areas in Thailand pointed out that mobile and industrial sources account for 42% to 57% and 15% to 44% of VOCs concentrations, respectively. He et al. [19] analyzed the composition characteristics and potential tracers of non-coal emission sources, including refueling, solvent use, and industrial and commercial activities. Wei et al. [20] collected samples near different production units in a large oil refinery in Beijing, showing that C_3_–C_6_ alkanes, C_3_–C_4_ alkenes, and aromatic hydrocarbons were the most important species.

With the sustained development of the automotive industry, the number of rubber tires has quietly increased, waste tire production will exceed 20 million tons in China, ranking first in the world, and will gradually become a new source of solid-waste pollution, and the problem of “black pollution” is severe. The latest research has shown that burning and energetic “recycling” in cement plants appear to be the dominant usages of old tires; feedstock recycling would be a very beneficial approach in the case of tires and rubber in general [21]. In China, tire recycling is mainly distributed in Hebei, Shandong, Shanxi, and other provinces; most of them are small and medium-sized enterprises, which have backward technology and treatment equipment, and when solving the pollution problem of the waste tire, a large number of complex organic waste gas will be emitted, so the environmental effects and health effects cannot be ignored.

At present, research into the rubber industry mainly focuses on the exhaust gas emission characteristics and terminal treatment technology of raw rubber material treatment [22,23,24,25]; there is less literature reporting on the composition and concentration of pollutants emitted by processes from recycled rubber plants [26]. Based on the above background, in this study, we selected VOCs from different processes in a typical recycled rubber plant in Shanxi Province for detection and analysis and then the emission characteristics of VOCs were studied deeply, the industry markers were determined, and the health risks were evaluated. Finally the L_OH_ was calculated to quantify the environmental effects of the industry. The research results help the government to understand the emission characteristics and risk levels of VOCs in the recycled rubber plant better and provide an essential reference for China to standardize the measurement method of VOCs source composition spectrum and build a localized source composition spectrum.

## 2. Materials and Methods

### 2.1. Description of the Waste Rubber Tire Recycling Process

There are many ways to recycle waste rubber tires. The principle is that the network structure of waste rubber is destroyed under the comprehensive action of heat, oxygen, mechanical force, and a chemical regeneration agent so that the plasticity of waste rubber can be restored and achieve regeneration [27,28]. Among them, the regeneration process, which uses a dynamic devulcanization tank and rubber mixing, is widely used in North China. In this study, we selected a representative recycled rubber plant in Shanxi province, which used waste tires as the main raw material. The plant had typical recycling processes that could be generally divided into two stages, i.e., rubber powder preparation and recycling.

The detailed processes are described as follows: The collected waste tires are centrally treated and then crushed step by step by the tire crushing equipment (crumb rubber production step), and the main particulate matter can be effectively removed by the gas collection system and the dust collector (more than 99%). After preparation, the crumb rubber and a certain proportion of chemical additives (coal tar, crude aromatic oil, cracking heavy oil, etc.) are reacted in the dynamic devulcanization tank at a high temperature (210~250 ℃) and high pressure (2.0~2.3 MPa), thereby plastic and viscous rubber aregenerated (devulcanization step), and then the devulcanized crumb rubber is transported through the conveyor belt to the cooling yard for cooling (cooling and yarding step), then the cooled crumb rubber enters the mill and refiner machine in turn and is continuously affected by the mechanical shear force of the rotor in the machine, the temperature was raised to about 160 ℃ (rubber refining step), which can improve the plasticity of the rubber powder to the level of replacing natural rubber; finally, it is pressed into a tablet. The typical process flowchart of waste rubber tire recycling is shown in Figure 1.

The contaminant emitted from the recycling is usually different from that emitted from crumb rubber production. The production process mainly emits particulate matter, which only involves physical processes, while the recycling process mainly emits organic waste gas (more than 90%) with high temperature and high discharges. The specific reasons for the occurrence of VOCs are expressed as follows: In the devulcanizing step, the production process of the dynamic devulcanization tank equipment is not continuous, and the exhaust gas needs to be discharged intermittently before unloading, which will engender a large amount of organic waste gas, the cooling and yarding step has just undergone a high-temperature state of chemical reaction, and the rubber powder is exposed to the workshop throughout the process; subsequently, a great deal of organic waste gas will be released, the high temperature of the rubber refining process causes low-molecular compounds to be volatilized quickly.

### 2.2. VOC Sampling and Analysis

In this study, five sampling sites, which could represent the emission characteristics of the processes were selected, including the dynamic devulcanization tank pressure relief port, screw cooling conveyor outlet, crumb rubber yard, mill, and refiner. When collecting samples, in order to ensure the accuracy of samples and reduce accidental errors, three parallel discharge samples were collected at each sampling site by a summa canister (Entech, Malvern, PA, USA) equipped with a restrictor valve (39-RS-x; Entech Instruments), the collection time of each sample was completed within 5–10 min, all of the collected samples were completed within 1 day. The VOC emissions during the one-day sampling period may not be fully representative of the annual emissions from the rubber plant. However, during the sampling period, each facility in the plant was in normal production, and the processes were operating normally, so sampling at this time had preferable representativeness. For the yard and cooling link, our fugitive emission sampling point was set outside the pollution source at 1 m; for the rest of the process with the gas collection hood, our sampling point was placed at 0.5m directly above the gas collection hood of each production facility. Table 1 shows specific information about sampling. A total of 15 samples from industrial sources were collected (including fugitive and organized emission samples). After sampling, all of the VOC samples were sent back to the laboratory within 20 days for preservation and analysis. Afterward, these samples were injected into the GC/MS-FID system (Agilent 7890A/5975C).

The detailed analysis process of the samples is described as follows: Before entering the GC/MS analysis, the collected samples were condensed and enriched, firstly by the three-stage low-temperature cold trap of the atmospheric pre-concentrator (Nutech 8900DS), and the sample was first passed through a primary cold trap composed of glass beads (at a temperature of −150 °C) to remove water vapor, then the VOCs in the samples were enriched by a secondary cold trap consisting of a Tenax adsorbent (at −20 °C) while removing nitrogen, oxygen, and carbon dioxide, the desorption temperature and time was 225 °C and 3 min, respectively, finally, the samples were condensed and concentrated by the third cold trap stage (at −165 °C) and transferred to the GC/MS system (Agilent 7890A/5975C) for analysis. During the analysis, high-purity helium gas was used as the carrier gas to load the collected air samples into a chromatographic column DB-1 (60 m × 0.32 mm × 1 µm, Agilent Technology, Santa Clara, CA, USA) for separation. Subsequently, the C_2_–C_3_ VOCs were separated on a PLOT-Q column (30 m × 0.32 mm × 20 µm, Agilent Technology, Santa Clara, CA, USA) and quantified using a flame ionization detector (FID), while the C_4_–C_12_ compounds (including hydrocarbons, oxygenated VOCs, halocarbons, and other species) were separated on a nonpolar column (50 cm × 0.15 mm (I.D)) and detected using quadrupole mass spectrometry detector (MSD). The oven temperature was initially held at 35 °C for 5 min, then increased to 150 °C at 5 °C/min for 7 min, and finally increased to 200 °C at 10 °C/min for 4 min, the ion source was electron ionization (EI), and the ion scan mode was selective ion scan. The target compound was identified according to the retention time of the sample and mass spectrum and quantified by the external standard curve method. A total of 100 VOC species were measured in this study, including 11 alkenes, 29 alkanes, 16 aromatics, 36 halocarbons, 6 ether esters, acetylene, and carbon disulfide. Standard curves were prepared by diluting a mixture of photochemical assessment monitoring stations (PAMS) and TO-15 (Spectra Gases, Linde, Stewartsville, NJ, USA) standard gases to corresponding concentrations (5, 10, 20, 50, and 100 ppbv), respectively. A replicate of every 10 samples was analyzed to ensure that the relative standard deviation was less than 30%.

### 2.3. Analytical Methods

#### 2.3.1. Coefficient of Divergence Method

The coefficient of divergence method can be used to analyze the similarity between VOC composition spectra in different processes [29]. The equation for calculating CD_ij_ is as follows:(1)$${\mathrm{CD}}_{\mathrm{jk}}=\sqrt{\frac{1}{p}}{\sum_{i=1}^{p}\left(\frac{X_{\mathrm{ij}}{-X}_{\mathrm{ik}}}{X_{\mathrm{ij}}{+X}_{\mathrm{ik}}}\right)}^{2}$$
where CD_jk_ is the divergence coefficient, j and k are different production processes, X_ij_ and X_ik_ are the content of component i in process j and k, respectively, and p is the number of chemical components involved in the calculation.

The CD values range from 0 to 1. The closer the CD is to 0, the stronger the similarity between the composition profiles, and the closer the CD is to 1 indicates that the content of each component between the composition profiles varies greatly. At present, there is no standard for dividing the similar grades clearly between the compositional profiles in domestic studies. When using this method, Wang et al. believe that two component spectra with divergence coefficients between 0 and 0.2 must be similar, two component spectra between 0.2 and 0.5 may be similar, and two component spectra between 0.5 and 1 must not be similar. This study analyzed the degree of similarity based on this criterion.

#### 2.3.2. Health Risk Assessment

Considering that the rubber regeneration process requires the manual operation of equipment and film trimming, operators are exposed to the unorganized exhaust gas in the factory for a long time, and the exposure to benzene and benzene series will cause potential harm to the nerves and immune system of the body [30]; therefore, the health risks of VOCs emitted from various processes in rubber recycling were evaluated in this study. Exposure to air pollutants by ingestion, inhalation, and dermal contact are vital exposure scenarios for people. In industrial parks, inhalation is generally considered to be the main route of exposure to VOCs for workers [31,32,33]. In this study, the lifetime cancer risk (LCR) and hazard index (HI) (https://www.epa.gov/risk/risk-assessment-guidance-superfund-rags-part-f) (accessed on 12 July 2022) were used to assess the carcinogenic and noncarcinogenic risks of inhaled VOCs for workers. The calculation can be expressed by the following equations:(2)$$\mathrm{LCR}=\frac{C_{i}\times \mathrm{ET}\times \mathrm{EF}\times \mathrm{ED}}{365\times {\mathrm{AT}}_{\mathrm{ca}}\times 24}\times \mathrm{IUR}$$
(3)$$\mathrm{HI}=\frac{C_{i}\times \mathrm{ET}\times \mathrm{EF}\times \mathrm{ED}}{365\times {\mathrm{AT}}_{\mathrm{nca}}\times 24}\times \frac{1}{\mathrm{RfC}}$$
where C_i_ represents the daily environmental concentrations of VOCs (μg/m^3^); ET, EF, and ED represent the daily exposure time (h·d^−1^), exposure frequency (d·a^−1^), and exposure duration(a) for workers successively, take 8 h, 250 d, and 20 a, respectively; AT_ca_ and AT_nca_ represent the average times (a) of carcinogenic and noncarcinogenic effects, take 70 a and 25 a, respectively. The selection of the above parameters refers to national standards and related research reports [34,35]. IUR represents the inhalation unit risk (μg/m^3^)^−1^ of VOCs species used for carcinogenic risk assessment, and RfC represents the reference concentration (μg/m^3^) of the VOC species used for noncarcinogenic risk assessment. IUR and RfC values are derived from the U.S. EPA (United States Environmental Protection Agency) (https://www.epa.gov/fera/risk-assessment-carcinogenic-effects) (accessed on 12 July 2022) and the U.S. EPA IRIS (United States Environmental Protection Agency Integrated Risk Information System (https://iris.epa.gov/AtoZ/?list_type=alpha) (accessed on 12 July 2022).

#### 2.3.3. OH Radical Loss Rate (L_OH_) Calculation Method

In existing studies, the photochemical O_3_ creation potential (POCP) method, the maximum incremental reactivity (MIR), and the OH-reactivity-based (L_OH_) method were created to investigate the contribution of individual VOC species to O_3_ formation [36,37,38]. In this study, the L_OH_ value among each process was calculated. The method used to calculate the L_OH_ of individual VOCs species is described below:(4)$$L_{i}{}_{\mathrm{OH}}{=K}_{i}{}_{\mathrm{OH}}\times {\left[\mathrm{VOC}\right]}_{i}$$
where L_iOH_ denotes the OH radical loss rate of VOC species i, [VOC]_i_ represents the concentration of VOC species i, K_iOH_ denotes the reaction coefficient of VOC species i with OH radicals, and the K_iOH_ values are from Atkinson (2003) [9]. The total L_OH_ of the VOCs are the sum of the each L_iOH_.

## 3. Results and Discussion

### 3.1. Total Concentrations and Component Characteristics of VOCs from Detailed Processes

Considering that the measured concentration of carbon disulfide is too low (0–2 µg/m^3^), its analysis will be ignored. The concentrations of the measured VOC species at various sampling sites for each step are summarized in Table 2. The monitoring data show that the concentration of total VOCs (TVOCs) ranged from 1000 ± 99 to 19,700 ± 19,000 µg/m^3^. The TVOCs concentrations for devulcanizing (19,700 ± 19,000 µg/m^3^) and cooling (10,500 ± 1600 µg/m^3^) were obviously higher than those in the open training (2800 ± 1500 µg/m^3^), yarding (2300 ± 340µg/m^3^), and refining processes (1000 ± 99 µg/m^3^), indicating assuredly that high-intensity VOC concentrations were related to different processes in the recycled-rubber plant. For devulcanizing, the highest concentration was monitored, which used a lot of chemical additives and had the highest temperature among all of the steps; therefore, the organic waste gas was released more. The concentration of alkanes, alkenes, halocarbons, aromatics, ether esters, and acetylene measured in each process are also shown in Table 1. We found that aromatics were detected at the highest level (500 ± 180–12,700 ± 10,400 µg/m^3^) in each process, and alkanes were the second group (120 ± 70–4700 ± 6500 µg/m^3^) in devulcanizing, cooling, yarding, and refining, except that in the open training, halocarbons (270 ± 280 µg/m^3^) were detected as the second component. The values of alkenes and halocarbons were also higher in the devulcanizing with 1200 ± 1900 and 960 ± 700 µg/m^3^. The remaining components as, ether esters, and acetylene were monitored with lower concentrations. The above analysis indicated that the main emission components of each process were shown to be quite similar, aromatics; and alkanes were the predominant components. Considering that the production process of recycled rubber was single, in addition to the use of chemical agents in the devulcanizing step from raw materials and finished products, others were mainly based on high temperature and physical forces.

Table 3 compares the dominant components emitted by different rubber industries, in which aromatics were pointed out as the main components in each study, which were consistent with the results of this study. In addition, Gagol et al. [39] measured sulfides as the principal emission in the reclaimed rubber process at 150 °C; Kamarulzaman et al. [24] determined the exhaust gas emission characteristics of the natural rubber drying process at 30 °C and 60 °C, and detected important components such as pinene; this may be due to the fact that both studies used a different analytical technique (dynamic headspace and gas chromatography-mass spectrometry) than this study, while the raw material measured in the latter study was natural rubber. Qianqian Li et al. [40] studied the characteristics of VOCs emitted in the three main process stages of rubber products and showed that the concentration of VOCs in the vulcanization stage was second only to the highest spray stage, alkanes and alkenes were the most important components; Huang et al. [41] measured dichloromethane as the major species in the mixing process, C_6_–C_8_ alkanes were dominant in the shaping process, and sulfides were released from the vulcanization process; Kwon et al. [26] pointed out that the main VOCs released during the heating process of waste rubber tires at 160 °C were alkanes, C_2_–C_4_ alkenes, etc. The above comparison results showed that marked regional differences occurred in the VOC emission characteristics of the recycled rubber plant, similarities and differences in VOCs emission characteristics were both found between different rubber industries, and the differences might be caused by the different product formulations, analytical techniques, and reaction conditions used in each rubber industry.

### 3.2. Key Species Compositions of VOC in Process

To analyse the emitted VOC in each process as a whole, the top ten species by concentration proportion for each process were selected, as shown in Figure 2, which accounted for about 68.08 ± 7.07%, 82.20 ± 1.12%, 90.17 ± 1.29%, 83.59 ± 3.65%, and 56.93 ± 7.91% of the TVOCs, respectively. We clearly found that aromatics (49.04 ± 15.20–89.79 ± 1.97%) were the richest component, and m/p-xylene (14.63 ± 4.07–48.87 ± 3.20%) provided the highest contribution, which could be possibly regarded as the VOCs emission marker from the recycled rubber plant. The “Compilation of Air Pollutant Emission Factors” (AP-42) and the Rubber Products Industry Pollutant Emission Standard (GB27632-2011) compiled by the State Environmental Protection Administration both point out that xylene is used as the emission factor, which is consistent with the results of this study. Furthermore, compared with yarding, 1,2,4-trimethylbenzene (5.08 ± 0.07–10.04 ± 4.59%), 1,3,5-trimethylbenzene (3.82 ± 0.08–5.04 ± 2.49%), dodecane (2.31 ± 0.46–7.45 ± 5.07%), 1,2,3-trimethylbenzene (2.24 ± 0.97–4.49 ± 2.47%), and 1-ethyl-3-methylbenzene (2.27 ± 0.11–3.08 ± 1.38%) were detected in a relatively richer proportion in the other four steps that exhibited similar emission characteristics. This result is reasonable for these four steps are carried out at obviously higher temperatures than yarding, and the boiling point of trimethylbenzene and dodecane are higher than benzene and toluene, so only relatively high proportions of toluene (16.78 ± 0.58%) and benzene (15.82 ± 0.39%) were detected in the yarding step. We also found that the proportions of styrene and ethylbenzene in both devulcanizing (4.12 ± 1.93%, 4.02 ± 1.96%) and cooling (3.74 ± 0.48%, 5.39 ± 0.63%) were bigger than other steps, styrene is widely used as raw materials for synthetic rubber.

As mentioned in Section 3.1, the TVOC concentrations in the devulcanizing and cooling were rather higher than those in the other processes. Between these two high-pollution processes, the main differences appeared in the alkenes, halocarbons, and alkanes. In the devulcanizing, the concentrations of alkenes, halocarbons, and alkanes were 10, 5, and four-times higher than those in cooling, respectively, in which n-butene (4.82 ± 3.20%), 2-methylpentane (3.73 ± 2.90%), carbon tetrachloride (1.75 ± 1.04%), and tetrachloroethane (1.68 ± 10.49%) provided higher percentages. For open training, tetrchloroethane (7.60 ± 5.78%) was observed as the second largest contributor, which is used as a non-combustible solvent for rubber in industry and is easy to maintain in rubber use. The main species in refining were dodecane (7.45 ± 5.07%), methylcyclohexane (3.62 ± 4.17%), and acetylene (2.85 ± 1.59%), for both the international standard classification and the Chinese standard classification have outlined that dodecane can be used as a chemical additive in rubber products. This result revealed that specific similarities and differences were found in the VOC emission characteristics of each process affected by raw materials, production conditions, and process equipment.

As noted above, although the main components of VOCs emitted by each process were similar, the content of different species varied greatly. In order to quantitatively compare the differences between the VOCs source profiles of different steps, we also used the coefficient of divergence method to analyze the similar degree, as illustrated in Table 4. Eight species were selected to participate in the calculation, which accounted for more than 5% of each process. We found that the divergence coefficients between the VOCs source profiles of different steps ranged from 0.308 to 0.654, among which the devulcanizing and open training were relatively similar with 0.308; the differences between the yarding and other processes were relatively higher, while the divergence coefficients were more than 0.5, the maximum was 0.654; the divergence coefficients between the other processes were relatively small, ranging from 0.2 to 0.5, which were judged to be possibly similar. This calculation was consistent with the previous analysis results and could be considered reasonable.

As mentioned in the above analysis, devulcanizing, cooling, open training, and refining were confirmed as the key steps of the recycled rubber plant in this study. Table 5 lists the comparison of TVOC concentrations and characteristic species by the rubber industry in this study and other typical industrial sources. The concentration of TVOCs in the recycling rubber process was significantly lower than that of the coking industry, petrochemical, and pharmaceutical industry. Comparing the main VOCs emitted by various industries, it was found that the main emissions of m/p-xylene, trimethylbenzene, and dodecane from recycling rubber production were significantly different from that emitted by solvent, petrochemical, coking, and pharmaceutical. In addition, the proportion of aromatics was also compared in these industries, which for the rubber plant (49.04–89.79%) was significantly higher than that of other industries except pharmaceutical (67.8–95.3%). Considering the health problems of workshop equipment operators, more attention should be paid to the discharge of pollutants from the rubber industry.

### 3.3. Health Risk Assessment among the Process

Considering the health problems of workers in a rubber factory, the health risks of the VOC species emitted by each process were assessed. In this study, according to the U.S. EPA IRIS database, only seven carcinogenic substances and 11 non-carcinogenic species were detected. The inhalation unit risk (IUR) and reference concentrations (RfC) values of these 18 VOCs are shown in Table 6. Due to carcinogenic VOCs also having non-carcinogenic risks, the carcinogenic risks of seven VOCs and the non-carcinogenic risks of 18 VOCs were evaluated in this study. Both carcinogenic and non-carcinogenic risk values are provided by US EPA. When the carcinogenic risk value LCR is less than 10^−6^, it is regarded as an acceptable risk level, and if it is between 10^−6^ and 10^−4^, indicating that there is a potential carcinogenic risk; when it is greater than 10^−4^, it is regarded as a large carcinogenic risk; when the carcinogenic risk value HI > 1, it indicates that there is a non-carcinogenic health risk, and when HI < 1, it indicates that the risk is negligible (https://www.epa.gov/sites/production/files/2015-09/documents/rags3adt_complete.pdf) (accessed on 12 July 2022).

In the whole process, the HI value exceeded the threshold limit value of 1, indicating that the exhaust gas emitted by the five steps in the park had non-carcinogenic health risks. Among them, the non-carcinogenic risk mean value of the devulcanizing is the highest at 24.71, followed by cooling, reaching 13.98; the values of HI in other steps were relatively low (1.18–4.84). Figure 3 only shows seven substances with a mean HI of non-carcinogenic health risk greater than 1, including 1,2,4-trimethylbenzene, 1,3,5-trimethylbenzene, 1,2,3-trimethylbenzene, benzene, o-xylene, trichloroethylene, and m/p-xylene. For devulcanizing, the HI values of these seven substances were all higher than 1, in which m/p-xylene had the largest value of 8.62 at the mean level. The HI values of m/p-xylene, 1,2,4-trimethylbenzene, and 1,3,5-trimethylbenzene in the cooling were greater than 1, with 8.58, 1.62, and 1.24, respectively. In the yarding, higher HI values were observed for benzene and m/p-xylene (2.25, 1.94). The HI value of m/p-xylene in the opening training was also greater than 1 with the mean value of 2.48, and the HI value of each species in the refining was less than 1, which was within the acceptable risk level. The above results showed that devulcanizing and cooling would cause greater non-carcinogenic health risks; m/p-xylene and benzene were the main contributors to the HI value of the plant.

The calculation results of the LCR value are shown in Figure 4, and the values of LCR in the devulcanizing, cooling, and yarding were more than 10^−4^, which were 5.7 × 10^−4^, 1.7 × 10^−4^, and 2.0 × 10^−4^, respectively, and were confirmed to be of relatively high carcinogenic risk. The mean LCR values of open training and refining were 7.0 × 10^−5^ and 2.8 × 10^−5^, respectively, indicating a potential carcinogenic risk. For devulcanizing, benzene, carbon tetrachloride, and ethylbenzene were the main contributors to the LCR values, with 3.0 × 10^−4^, 1.3 × 10^−4^, and 1.3 × 10^−4^, respectively; meanwhile, trichloroethylene and 1,3-dichlorobenzene had LCR values between 10^−6^ and 10^−4^. Benzene (6.0 × 10^−5^), ethylbenzene (9.2 × 10^−5^), carbon tetrachloride (1.0 × 10^−5^), and 1,3-dichlorobenzene (2.6 × 10^−6^) in cooling were potential carcinogenic risks. The LCR value of benzene in yarding was higher than 1 × 10^−4^, and that of 1,3-butadiene, ethylbenzene, and carbon tetrachloride were between 10^−6^ and 10^−4^. Both open training and refining had the same characteristic VOC species with LCR values, such as benzene (5.8 × 10^−5^, 9.6 × 10^−6^), ethylbenzene (7.5 × 10^−6^, 1.6 × 10^−6^), and carbon tetrachloride (2.9 × 10^−6^, 1.6 × 10^−5^). T compounds not mentioned in any of the above analyses had LCR values below 10^−6^ and were considered to be within acceptable risk levels. Therefore, devulcanizing, cooling, and yarding in the plant had greater carcinogenic risks, and the others had potential carcinogenic risks. Benzene, ethylbenzene, and carbon tetrachloride were the dominant risk compounds that should be firstly considered to protect the health of workers in the recycled rubber plant.

However, due to the exposure parameters, such as IUR, ED, EF, ET, and AT being variable, large uncertainties still exist. These VOC levels were detected over a short time, whereas the assessment was defined as assuming a 25-year and 70-year lifespan of continuous exposure. Additionally, in this study, only inhalation risks were considered, possibly resulting in inaccurate risk assessments.

### 3.4. OH Radical Loss Rate

As is known to all, VOCs are the momentous precursors of ozone, and the OH loss rates (L_OH_) and ozone formation potential are widely used to assess ozone contribution. In this study, OH radical loss rate method is used to analyze the ozone contribution in different steps in order to evaluate the environmental effects of the rubber plant. Table 7 shows the L_OH_ evaluation results of VOC components. The L_OH_ values of VOCs emitted from different processes were found in the following decreasing order: devulcanizing > cooling > open training > yarding > refining. Devulcanizing and cooling were the largest and second-largest contributors to ozone, accounting for 56.56% and 29.19%, for the superior concentrations of aromatics and alkenes, which had the stronger propensity to ozone formation. Other steps (open training, yarding, and refining) had the lower L_OH_ values due to the low levels of active constituents, such as aromatics and alkenes, accounting for 7.04%, 4.84%, and 2.37%, respectively. The L_OH_ contribution characteristics of different kinds of VOCs were also displayed in Table 7. The clear result was that aromatics were the most abundant component in the whole plants, with the percentage ranging from 72.11% to 93.74%, followed by alkenes (2.81–20.62%) and alkanes (1.69–12.97%). Distinctly, it is essential to focus on the emission of aromatics.

For this reason, in this study, we conducted a further detailed analysis of the high-contribution species of L_OH_ in aromatics. The contribution proportions (%) for individual species in aromatics with their homologous L_OH_ values are shown in Figure 5. Dimethylbenzene (m/p-xylene and o-xylene), trimethylbenzene (1,3,5-trimethylbenzene, 1,2,4-trimethylbenzene and 1,2,3-trimethylbenzene), and styrene were found to be the main dominant species for almost all processes, accounting for 87.76%, 87.99%, 88.30%, 92.38%, and 89.26%, respectively. Among them, the most key species for L_OH_ were m/p-xylene (25.26–67.87%). Therefore, for the recycled rubber plant in this study, the emission level of aromatics, especially m/p-xylene, should be prioritized for control through L_OH_ activity evaluation. Further work is required for further in-depth study of the mechanism of ozone formation in recycled rubber plants.

## 4. Conclusions

A total of 15 samples from the typical recycled rubber plant were collected by summa canisters in a typical recycled rubber plant, and 100 VOC species were detected by the GC/MS-FID system. We observed that aromatics and alkanes were the predominant components in the whole process, followed by halocarbons and alkene. The result of key species compositions of VOC in the process showed that m/p-xylene could be possibly regarded as a VOC emission marker. We also found that specific similarities and differences in VOC emission characteristics in each process were affected by raw materials, production conditions, and process equipment; n-butene provided higher percentages in devulcanizing, for open training, tetrchloroethane was observed as the second largest contributor, dodecane, methylcyclohexane, and acetylene were the main species in refining, these were also confirmed by the coefficient of divergence method. The assessments of health risks showed that the non-carcinogenic risk value of the devulcanizing is the highest, the values of LCR in the devulcanizing, cooling, and yarding were more than 10^−4^ and were confirmed to be of relatively high carcinogenic risk, and open training and refining had potential carcinogenic risks. In the plant, m/p-xylene and benzene were the main non-carcinogenic species, while benzene, ethylbenzene, and carbon tetrachloride were the dominant risk compounds. In the evaluation results of L_OH_, devulcanizing and cooling steps were the largest and second largest contributor, and aromatics were the most abundant component to ozone formation, in which m/p-xylene was the most key individual species and should be prioritized for control. The key species detected in this study provide important information for the formulation of emission reduction policies.

## Figures and Tables

**Figure 1 ijerph-19-08753-f001:**
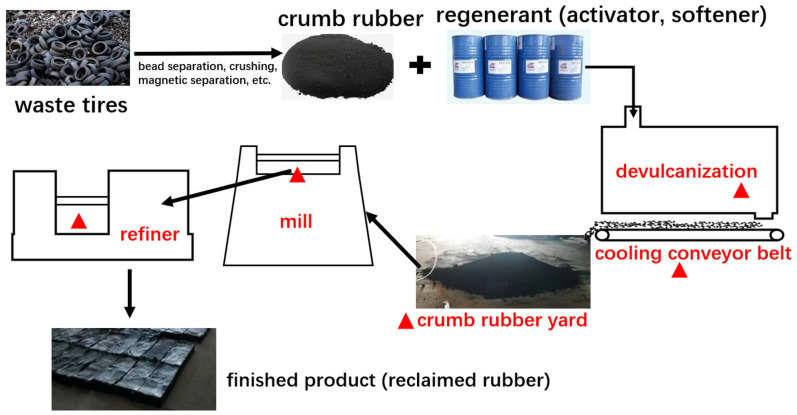
Typical process flowchart of waste rubber tire recycling.

**Figure 2 ijerph-19-08753-f002:**
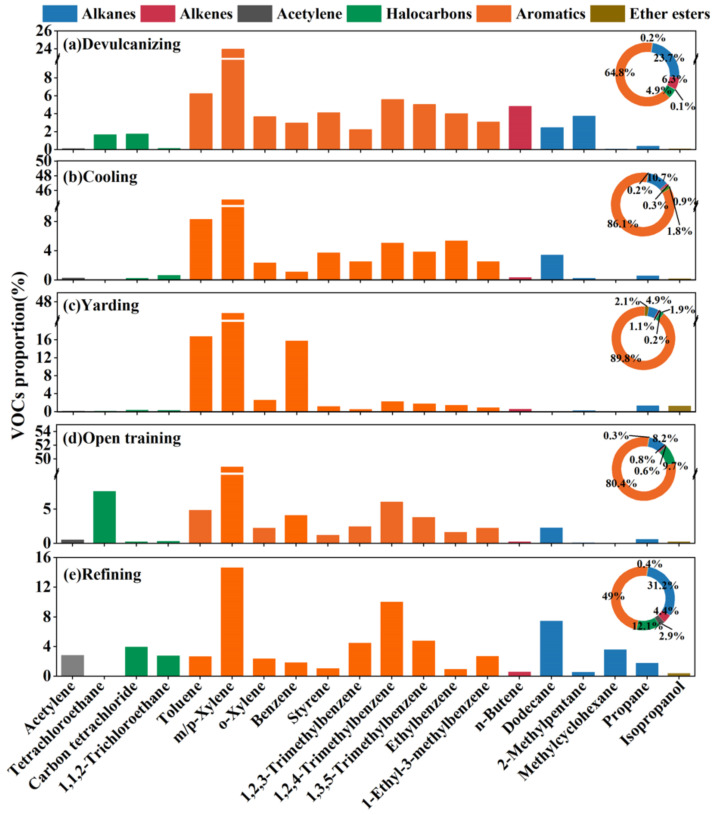
VOCs source profiles for the recycling plant Note: species included in the bar graph were in the top ten of the total concentrations of each profile, while the ring charts contain all measured species.

**Figure 3 ijerph-19-08753-f003:**
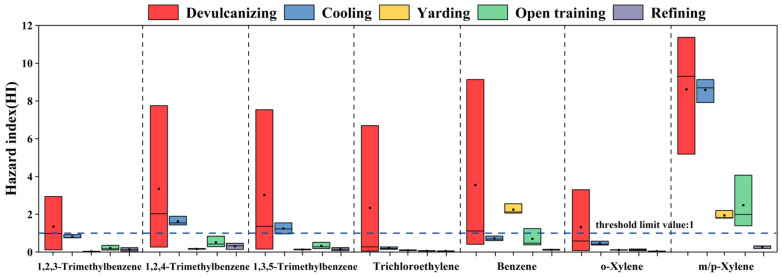
HI value of compounds from each process. The box represented the 25–75th percentiles of HI values. The middle square and middle line represented the mean and the median values of HI values, respectively.

**Figure 4 ijerph-19-08753-f004:**
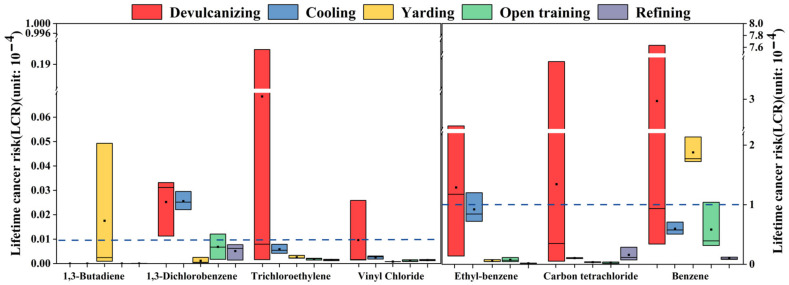
LCR value of compounds from each process. The box represented the 25–75th percentiles of LCR values. The middle square and middle line represented the mean and the median values of LCR values, respectively.

**Figure 5 ijerph-19-08753-f005:**
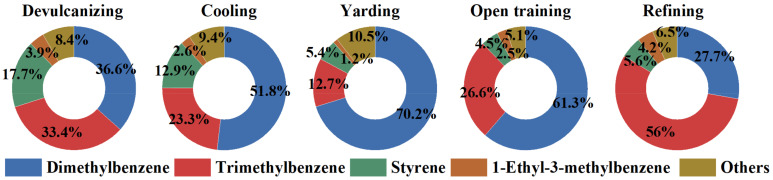
The L_OH_ contribution of aromatics in each process.

**Table 1 ijerph-19-08753-t001:** Sampling information in each measured sample (µg/m^3^).

No. of Samples	Sampling Objective	Sampling Site	Emission Form	Facility Condition
1	Dynamic devulcanization tank pressure relief port	0.5 m above	Organized	Normal
2	Screw cooling conveyor outlet	1 m horizontal	Fugitive	Normal
3	Crumb rubber yard	1 m horizontal	Fugitive	Normal
4	Mill	0.5 m above	Organized	Normal
5	Refiner	0.5 m above	Organized	Normal

**Table 2 ijerph-19-08753-t002:** Concentration characteristics of VOCs components in each process (µg/m^3^).

Process	Devulcanizing	Cooling	Yarding	Open Training	Refining
Alkanes	4700 ± 6500	1100 ± 200	120 ± 70	230 ± 100	320 ± 80
Alkenes	1200 ± 1900	100 ± 10	30 ± 5	20 ± 4	50 ± 10
Halocarbons	960 ± 700	190 ± 70	50 ± 12	270 ± 280	120 ± 70
Aromatics	12,700 ± 10,400	9000 ± 1300	2100 ± 257	2200 ± 1100	500 ± 180
Ether esters	30 ± 35	20 ± 6	50 ± 8	7 ± 10	5 ± 2
Acetylene	20 ± 8	30 ± 1	4 ± 0	20 ± 0	30 ± 10
TVOCs	19,700 ± 19,000	10,500 ± 1600	2300 ± 340	2800 ± 1500	1000 ± 99

**Table 3 ijerph-19-08753-t003:** Comparison of key VOCs components in other rubber industries.

References	Characteristic Components of the Main Process
The study	Aromatics, Alkanes
Huang et al., (2022) [41]	Dichloromethane, C_6_–C_8_ alkanes, Sulfides, Aromatics
Kamarulzaman et al. (2019) [24]	Alkanes, Aromatics, Pinenes
Qianqian Li et al. (2019) [40]	Alkylene, Aromatics, Sulfides
Gagol et al. (2015) [39]	Aromatics, Sulfides
Kwon et al. (2015) [26]	Akanes, C_2_–C_4_ alkenes, Aromatics

**Table 4 ijerph-19-08753-t004:** Divergence coefficients between the VOCs source profiles in each process.

Process	Devulcanizing	Cooling	Yarding	Open Training	Refining
Devulcanizing	0				
Cooling	0.416	0			
Yarding	0.615	0.654	0		
Open training	0.308	0.465	0.595	0	
Refining	0.489	0.539	0.654	0.490	0

**Table 5 ijerph-19-08753-t005:** Comparison with VOCs emitted by other typical industries.

Emission Source	Concentration of TVOCs	Major Species	The Proportion of Aromatics
This study	1000–19,700 µg/m^3^	m/p-Xylene, Trimethylbenzene, Dodecane	49.04–89.79%
Solvent [42]	--	Ethylene, Undecane, Benzene, Ethylbenzene, m/p-Xylene	12.91–47.48%
Petrochemical [43]	99.5–95,253.0 µg/m^3^	Undecane, Benzene, Toluene, n-pentane, cis-2-butene	2.8–60.0%
Coking [44]	690.29–62,651.59 µg/m^3^	Ethylene, Ethane, Benzene, Toluene, Naphthalene	12.93–93.81%
Pharmaceutical [45]	827–33,700 µg/m^3^	Toluene, Dichloromethane, Ethanol, Methanol	67.8–95.3%

**Table 6 ijerph-19-08753-t006:** VOCs risk assessment related parameters.

Compounds	IUR (μg/m^3^)^−1^	RfC (μg/m^3^)	Compounds	IUR (μg/m^3^)^−1^	RfC (μg/m^3^)
n-hexane		700	1,2,4-Trimethylbenzene		60
Cyclohexane		6000	1,2,3-Trimethylbenzene		60
1,3-Butadiene	0.00003	2	Trichloroethylene	0.0000041	2
Benzene	0.0000078	30	Chlorobenzene		1000
Toluene		5000	1,2,4-Trichlorobenzene		200
Ethylbenzene	0.0000025	1000	1,3,5-Trimethylbenzene		60
m/p-xylene		100	Vinyl chloride	0.0000088	100
O-xylene		100	Carbon tetrachloride	0.000006	100
Styrene		1000	1,3-Dichlorobenzene	0.000004	20

**Table 7 ijerph-19-08753-t007:** The L_OH_ contribution of VOCs from all the recycling processes (s^−1^).

Process	Alkanes	Alkenes	Aromatics	L_OH_
Devulcanizing	250 ± 340	720 ± 1100	2500 ± 2200	3500 ± 3700
Cooling	70 ± 16	56 ± 6	1700 ± 240	1800 ± 260
Yarding	5 ± 2	14 ± 2	280 ± 34	300 ± 40
Open training	15 ± 8	12 ± 2	410 ± 220	430 ± 230
Refining	20 ± 4	22 ± 4	110 ± 46	150 ± 44

## Data Availability

Data are available from the authors at reasonable written request after authorization by the School of Environmental Science and Engineering, Taiyuan University of Science and Technology, China.

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
