# Peer review of "Emission Characteristics and Health Risks of Volatile Organic Compounds (VOCs) Measured in a Typical Recycled Rubber Plant in China"

_ijerph, 2022, doi:10.3390/ijerph19148753_

Round 1
Reviewer 1 Report
All abbreviations should be explained. Especially in the Abstract the authors use the abbreviation TVOC, which is not explained not only in this part but also in the others. Another abbreviations: PID, EI, etc….
Numerical values are given along with errors. Eg. "The TVOCs concentration ranged from 1019.90±99.01 to 19672.43±19099.63μg/m3”. Are the errors given standard deviations? How were the above statistical error values calculated?
The authors analysed data collected from Hebei, Shandong, Shanxi and other provinces. In my opinion, the title of the publication should be changed. It should say that it is data collected in China, e.g. Emission characteristics and health risks of VOCs measured in a typical recycled rubber plant in China.
Lines 194-196 Please systematise the units, either “h day-1” or “d•a-1”.
Please explain what "a" means in the above unit.
Reviewer 2 Report
The manuscript describes the results from analyses of a small set of samples taken within industrial rubber recycling plants. Unfortunately the study lacks context, and ends up over-interpreting the measurements. Numerous updates are more detailed explanations are needed before the study can be acceptable for publication.
Detailed comments:
1. In several places in the manuscript, the measured concentrations are labeled as "emissions", which is incorrect. Neither are concentrations equal to emissions, nor can concentrations act as emission proxies in most cases.
2. The descriptions of sampling are insufficient. Since the study has very few samples, only 15 in total, a detailed list of all samples needs to be provided, including times of sampling, nearness to (which?) sources, a brief explanation of representativeness of each sample (number of samples per source category), and the time it took to collect them.
3. The analytical method appears to use two columns in series. It should be described in a bit more detail, especially the transfer and preconcentration steps.
4. Data is given at a precision that makes no sense, namely a variability several order of magnitude higher than the number of given digits. Thus, all data ought to be rounded to significance levels, e.g. "10491.17+-1559.58" to 10491+-1559 or even 10500+-1600.
5. LCRs and HIs only make sense when calculated for representative samples, i.e. samples that actually reflect exposure. This has not been demonstrated.
6. Similarly, OFPs only make sense when the samples reflect typical ambient concentrations. This seems clearly not the case, as the samples were taken to reflect typical emissions compositions, and thus this section ought to be dropped. A verbal description using OH reactivity values is entirely sufficient to demonstrate potential impacts on local ozone formation.
7. If the US-EPA AP-42 database mentioned in the text actually contains emission estimates for this industry, the given compositions (if any) of those emissions should be compared to your results.
8. If possible, the (known or assumed) composition of the "regenerant" should be provided so that the reader gets a better idea of the origin of the emitted hydrocarbons as compared to the (recycled) tire rubber waste itself. Same is true for other chemicals used.
9. Section 4 does not contain conclusions, but reads like a summary.
10. Minor issues: there are several awkward wordings throughout the text (e.g. "breezily", "fleetly", "prioritily") that need to be fixed. Abbreviations need to be identified upon first usage. Special names, like "sostenuto development" need an explanation.
Reviewer 3 Report
The paper is very well written and organized. Figures and tables are informative. However, many points should be illustrated before final publication.
1- VOCs full name should be mentioned in the title and abstract. Moreover, OFP full name should be mentioned in the title.
2- Short description about methodology should be mentioned in the abstract before results.
3- The following article should be cited and included in the discussion https://www.sciencedirect.com/science/article/pii/S001393512200812X
https://www.sciencedirect.com/science/article/pii/S0304389419304923?via%3Dihub
4- Abbreviation list should be added.
5- The most important point, why the authors did not used GC-MS technique in addition to theoretical analytical methods? . in conclusion the authors stated “100 VOC species were detected by the 433 GC-MS/FID system” while this is not mentioned in methods. This is very confusing.
6- Conclusion should be more precise without numerical values. Numerical values should be summarized in abstract-( results section)
Best wishes
Round 2
Reviewer 2 Report
I thank the authors for their revision. Aside from being minor, you appeared to have done the revision hastily and at least one instance created new confusion.
First, a few comments using the point numbering in the authors' rebuttal:
2. Since this is a fairly new process evaluation, I do not consider the brief description of sampling sites and conditions adequate, meaning the authors' response is inadequate in my opinion. It is vital, and good analytical practice, to provide a clear and detailed description of when and how samples were taken in such a setting; not only because the number of samples was low (3 replicates per site), but also because this is not an established setting in a well-known environment of a widely practiced industrial procedure. It appears to me you may be providing a benchmark for future evaluations, and thus your results need to withstand scrutiny.
I therefore maintain that a closer description should be given other than "0.5 m above the production facilities". The circumstances need to be clear (e.g. enclosed building/facility vs. open air; samples taken during a scheduled or random release event from a port, or during a random time of day while the facility is operating? samples taken on the same day at different times? or during random visits to the facility? was facility operating normally? was is normal?), so need be the sampling periods ("instantaneous" meaning what? canister was filled within seconds by opening inlet valve? or air was collected over several hours using a restrictor?).
The required table would list the five locations and provide the required detail alongside any relevant environmental parameters during sample taking (i.e. the "circumstances" mentioned above)
3. The new sample GC analysis description now lists a "primary cold trap" (line 145/146) that supposedly removed "water vapor, nitrogen, oxygen and carbon dioxide". If that were the case, it would also have removed all VOCs in that step, so this cannot be correct. A cold trap typically operates at slightly below zero to remove water vapor, not O2, N2 and CO2, before the sample is enriched on a cooled adsorbent (here: Tenax). The method should list at what temperature the Tenax was desorbed under Helium flow, and for how long, before further focusing onto the final cold trap (from which injection occurred how?).
In addition, it is unclear to me what a "glass column" is (line 154/155). Is this simply a piece of a mid-bore (0.32 mm or narrow-bore?) transfer line to the MSD while the split entered the PLOT-Q column?
I strongly encourage you to read other researchers brief description of their analytical methods more closely, because I think this may be an English language use issue here with your own description.
5. Without the detailed description required in response to point #2, your assurances remain unsubstantiated.
8. see below
10: "sustenuto" is a musical term; you probably meant to say "sustained"
Additional comments:
For your paper to have any impact, you ought to carefully select your terminology, and use existing terminology from the field. Most importantly, I think, you ought to describe that the currently dominant tire rubber recycling is NOT done via the facilities you studied; burning and energetic "recycling" in cement plants appear to be the dominant usages of old tires (if this is different in China, say so). The next most common usage is crumb rubber which may be the same product as the "tire powder" in your manuscript, and if so, you ought to use that language. Next, "desulfurization" I think is also not the correct terminology in use either, "devulcanization" is. (Note that you mixed "former" and "latter" in lines 114/115 as the "rubber powder preparating" (aka crumb rubber production), the step listed second in line 114, is the "latter" step that "only involve [sic] physical processes" and "emits particulate matter")
There is an open access paper from 2020 here that you missed:
https://www.mdpi.com/1996-1944/13/5/1246/htm
which probably has more detail you ought to include with respect to the recycling process you investigated for its emissions.
Once you have put your work into this context better, you can hope for some relevance since you showed important VOC emissions.
Reviewer 3 Report
the authors did all required changes , thanks
Author Response
Thank you again for your valuable advice to me.